# Spatiotemporal filtering method for detecting kinematic waves in a connected environment

**Eui-Jin Kim**[1], **Dong-Kyu Kim**[2☯], **Seung-Young Kho**[2☯], **Koohong Chung**[3]*

**1** Department of Civil and Environmental Engineering, Seoul National University, Seoul, Republic of Korea,
**2** Department of Civil and Environmental Engineering and Institute of Construction and Environmental
Engineering, Seoul National University, Seoul, Republic of Korea, **3** School of Civil, Environmental and
Architectural Engineering, Korea University, Seongbuk-gu, Seoul, Republic of Korea

☯ These authors contributed equally to this work.
* koohong@korea.ac.kr

doi.org/10.1371/journal.pone.0244329

**Data Availability Statement:** All relevant data are
within the paper and its Supporting Information
files.

**Funding:** Dong-Kyu Kim was supported by the
Basic Science Research Program through the
National Research Foundation of Korea (NRF)

## Abstract

Backward-moving kinematic waves (KWs) (e.g., stop-and-go traffic conditions and a shock
wave) cause unsafe driving conditions, decreases in the capacities of freeways, and
increased travel time. In this paper, a sequential filtering method is proposed to detect KWs
using data collected in a connected environment, which can aid in developing a traffic con-
trol strategy for connected vehicles to stop or dampen the propagation of these KWs. The
proposed method filters out random fluctuation in the data using ensemble empirical mode
decomposition that considers the spectral features of KWs. Then, the spatial movements of
KWs are considered using cross-correlation to identify potential candidate KWs. Asynchro-
nous changes in the denoised flow and speed are used to evaluate candidate KWs using
logistic regression to identify the KWs from localized reductions in speed that are not propa-
gated upstream. The findings from an empirical evaluation of the proposed method showed
strong promise for detecting KWs using data in a connected environment, even at 30% of
the market penetration rates. This paper also addresses how data resolution of the con-
nected environment affects the performance in detecting KWs.

## Introduction

Backward-moving kinematic waves (KWs) signal changes in traffic conditions (i.e., traffic
shockwaves [1] and stop-and-go traffic conditions [2, 3]) such that detecting traffic conditions
leading to the emanation of KWs and monitoring their propagations have become important
topics of research. The information gleaned from KWs is vitally important in the real-time
traffic management [4], in the evaluation of traffic safety [5, 6], and connected vehicle control
strategies [7–11]. However, monitoring the propagation of KWs based on conventional loop
detectors limits their usage only on roadways equipped with loop detectors, and existing pro-
cedures for detecting KWs also result in many false positives (FPs) and false negatives (FNs)
because the stochastic nature of the KWs is not considered.

For the reasons stated above, we developed a procedure for detecting and monitoring the
propagation of KWs based on data collected from connected vehicles (CVs). The proposed

funded by the Ministry of Science and ICT No. 2020R1F1A1074395.(https://www.nrf.re.kr/eng/index) The funders had no role in study design, data collection and analysis, decision to publish, or preparation of the manuscript. There was no additional external funding received for this study.

**Competing interests:** The authors have declared that no competing interests exist.

method considers the stochastic nature of the KWs to improve the detection performance. The proposed method filters out random fluctuations embedded in the data using the temporal and spatial attributes of KWs. This paper reports on the findings from an empirical evaluation of the proposed method based on different types of data (i.e., loop detector and CV), CV penetration rates, and data resolution. The findings show strong promise for detecting KWs using data in the CV environment, even at a market penetration rate of 30%. The impacts of the type of data and the resolution of performance suggest how the data should be collected to detect KWs.

The remainder of this paper is organized as follows. First, we discuss the relevant findings from some previous studies. The following section describes the site where the study was conducted and the data that were obtained. The next section provides a detailed description of the proposed method. Subsequently, we report the findings based on the evaluation of the proposed method. Brief concluding remarks and future research plans are provided at the end of the paper.

## Background

### Detecting kinematic waves in a connected environment

Connected vehicle technology (CVT) can provide real-time information to CVs through the vehicle-to-vehicle (V2V) and vehicle-to-infrastructure (V2I) communication systems [12]. The trajectories of individual vehicles can be collected in CV's on-board units (OBUs) at specific time intervals. The data from the CVs can be transmitted to roadside units (RSUs) that can send the data to the Traffic Management Center (TMC) to be processed further to derive additional information about existing traffic conditions [13]. Compared with historical, infrastructure-based sensors (e.g., loop detectors and video cameras) that generally collect the aggregated traffic data along with the specific roadway locations, the CV-based technology can collect data that have a finer spatiotemporal resolution (i.e., densely-spaced and high-frequency data) wherever CVs travel [14].

Recent studies [7, 8, 11, 14] evaluated the potential for controlling the speed of a platoon of vehicles to dampen the adverse impact of KWs. The intent of that approach was to prevent the formation of KWs and stop their propagation by creating more uniform traffic flow. Accurate detection of KWs was an essential part of the aforementioned studies since it dictates the false positive and negative rates of the traffic control strategies [7, 10]. However, providing reliable traffic information based on V2V and V2I in a CV environment can be challenging since it requires a sufficient market penetration rate of the CV, which had yet to be attained when this paper was written. In addition, the existence of physical barriers (i.e., trees and concrete walls), frequency interferences from different sources, and the fact that the data are stochastic in nature result in both the contamination and the loss of information [14, 15]. Therefore, filtering out the random fluctuations in traffic data is an important step in developing a method that can accurately detect KWs. In addition, the type of data (i.e., the data collected from the loop detector or the CV environment) and the temporal aggregation (i.e., the time interval during which traffic data are aggregated) affect the level of the random fluctuations that are filtered and the accuracy of detecting KWs. Thus, the performance of the proposed method was evaluated in this paper during different market penetration rates by considering various types of data and considering the temporal aggregation levels of data.

### Existing techniques for detecting kinematic waves

Previous empirical studies have reported that the propagation speed of backward- moving KWs typically ranges between 10 and 24 km/h, irrespective of whether the KWs were triggered by lane changes or other contributing factors. KWs caused by stop-and-go traffic typically display the oscillatory pattern with a period of 5–15 minutes [16–19]. Several different techniques have been used to extract the spatiotemporal attributes of KWs that are needed to evaluate and control traffic.

Subtracting the moving averages from the oblique cumulative vehicle count curves amplifies the temporal changes, and the propagations of KWs can be tracked visually [2, 3]. However, such procedures require the manual determination of the parameters in each road segment, which makes the systematic implementation of the method challenging. The periodic oscillations in the stationary time series can be amplified using Fourier transform (FT), but the FT can distort the signals out of the dominant period defined by the filtering window size. To address this problem, Li et al. proposed a short-time Fourier transform (STFT) to improve the measurement of the periodicity and amplitude of the KWs observed during the stationary time interval [18]. More recently, Zhao et al. applied the windowing concepts of STFT to the standard spectral envelope method [19]. Although these methods based on STFT can detect KWs propagated across space, the stationary time interval must be determined before the method can be applied. Treiber and Kesting proposed a multiple cross-correlation technique that measured the similarity of speed patterns from sequential detector pairs to trace groups of KWs [20]. Although they were able to estimate the spatiotemporal features of KWs, their procedures were not designed to detect individual KWs. The above studies focused on investigating spatiotemporal attributes of KWs based on the aggregated behavior of the KWs, rather than detecting individual KW to specify their location and time. Also, their methods commonly require the manual determination of the parameters such as background flow in each road segment [2] and stationary time interval [18].

Other researchers have been proposed methods that focus on detecting KW, rather than investigating attributes of the KWs. Zheng et al. used wavelet transformation (WT) to detect the abrupt changes in the average wavelet-based energy accompanied by the emanation of KWs [17]. This approach can be used to detect KWs systematically by capturing and amplifying the spectral and temporal features in different frequency bands and positions. The authors reported that WT can detect KWs using the individual trajectories of all of the vehicles in a given section of the roadway. However, such complete individual trajectory data (i.e., the individual trajectories of all of the vehicles that KWs pass through) often are not readily available [21]. Also, the result of the analysis was sensitive to the choice of the wavelet function (i.e., parameters for the method) [22]. Elfar et al. proposed a method to detect KW using the speed standard deviation (SSD) value of individual vehicles in a connected environment [10]. Elfar et al. reported that a method based on SSD can reduce the false-negative rate (i.e., not detecting KWs) in a fully (i.e., 100 percent CV penetration rate) and partial (i.e., less than 100 percent CV penetration rate) connected environment since the SSD are sensitive to reductions in speed. The performance of the above-mentioned studies was evaluated based on visual inspection of the vehicle trajectory data without considering false positives from the random fluctuations. Therefore, the robustness of these methods against random fluctuations in the data had not been evaluated. Such random fluctuations in practice can result in high false positives in detecting KWs.

Several issues for detecting KWs can be drawn from the above review of previous studies. First, a data-driven method for detecting KWs needs to be implemented without local calibration of the parameter. This generalizable procedure is an important factor in enabling it to be used in practice. Second, previous studies focus on amplifying the signal of interests, rather than removing random fluctuations that cause false positives in detecting KWs. The impact of random fluctuation would be more prominent in the data from CVT that is potentially contaminated by physical barriers and frequency interference of the data collection system. Last, previous studies conducted a visual inspection for evaluating detecting performance without quantitative measures. Quantitative evaluation is essential for implementing the KW detection method in practice. As a remedy, this paper proposes a data-driven method for detecting KWs. While the methods mentioned above dealt with random fluctuations in the data by amplifying the signal of interest using different window sizes, our proposed method focuses on removing random fluctuations using a sequence of steps. All the parameters in each step can be calibrated

only from the data without the subject judgment of the researcher. To evaluate the applicability of the proposed method in practice, we evaluated our method using quantified measures under different CV penetration and data resolution. More details are explained in the later sections.

## Study site and data

### Observed features of kinematic waves

The Next-Generation Simulation (NGSIM) project [23] collected data of the trajectories of individual vehicles in all of the lanes along the 640-m section of the freeway shown on the left side of Fig 1A. The vehicle trajectory data were extracted by automated vehicle tracking system based on video analysis technique [21, 24] using videos from several synchronized cameras [25]. The data were collected on June 15, 2005 from 7:50 to 8:35 A.M., and the data included the trajectories of 6,101 vehicles. Four lanes from the far left were used for this study and those data were uploaded as the supporting information in S1 Dataset. The individual vehicle trajectories in the passing lane (i.e., the far left lane) are shown on the right side of Fig 1A. The slopes of the individual vehicle trajectories indicate the speeds of the specific vehicles, i.e., the steeper the slope, the faster the vehicle was traveling. Different colors were used to display more clearly the changes in the speeds that were observed as the vehicles encountered the backward-moving KWs, which are marked by the two slanted solid black lines labeled $KW_1$ –$KW_6$. The width of KW ($W_1$) was dictated by the horizontal space between the two slanted lines.

The reduction in speed observed downstream did not always accompany the emanation of KWs, and Fig 1A provides evidence of the localized reductions in speed (LS). $LS_1$ to $LS_3$, which are marked near location D, denote the time period when the reduction in speed was observed at location D, while $LS_1$ to $LS_3$ are not propagated to location U. KWs were emanated sometime after the LS as in $KW_1$, $KW_2$, and $KW_3$, while $KW_4$ to $KW_6$ were not preceded by LS. LS causes changes in speed without the accompanying synchronized changes in flow. Such asynchronous changes can introduce inaccuracies in determining the location and time of the backward-moving KWs, and this can increase the false-positive rates of procedures that were intended to detect the emanation of KWs solely based on the reduction in the speed.

In an effort to analyze traffic conditions related to LS and KW systematically, virtual detectors that mimic the dual loop detector were placed at locations D and U to collect 5-second aggregated speed and flow data, as shown in the black boxes in Fig 1A; time intervals shorter than 5 seconds did not provide additional information compared with the 5-second aggregated data. Fig 1B shows the vehicle accumulation, i.e., the number of vehicles between D and U as a function of time, $L(t)$. Traffic density can be obtained directly by dividing the vehicle accumulation by the distance between D and U (210 m). The rescaled cumulative speed and flow at locations D and U are constructed using virtual loop detectors, as shown in Fig 1C–1F, labeled as $S_D$, $N_D$, $S_U$, and $N_U$, respectively. These curves amplify the temporal changes using the oblique coordinate, which is defined by the background speed, $\mu_0$, and flow, $q_0$, as shown in Eqs (1) and (2) [26].

$$N(t) = \sum_{t_0}^{t} n(t) - q_0 \tag{1}$$

$$S(t) = \sum_{t_0}^{t} s(t) - \mu_0 \tag{2}$$

where $N(t)$ and $S(t)$ are rescaled cumulative flow and speed from $t_0$ to $t$, respectively; $n(t)$ and $s(t)$ are aggregated flow and speed at time $t$, respectively. The changes in slope shown in Fig 1C and

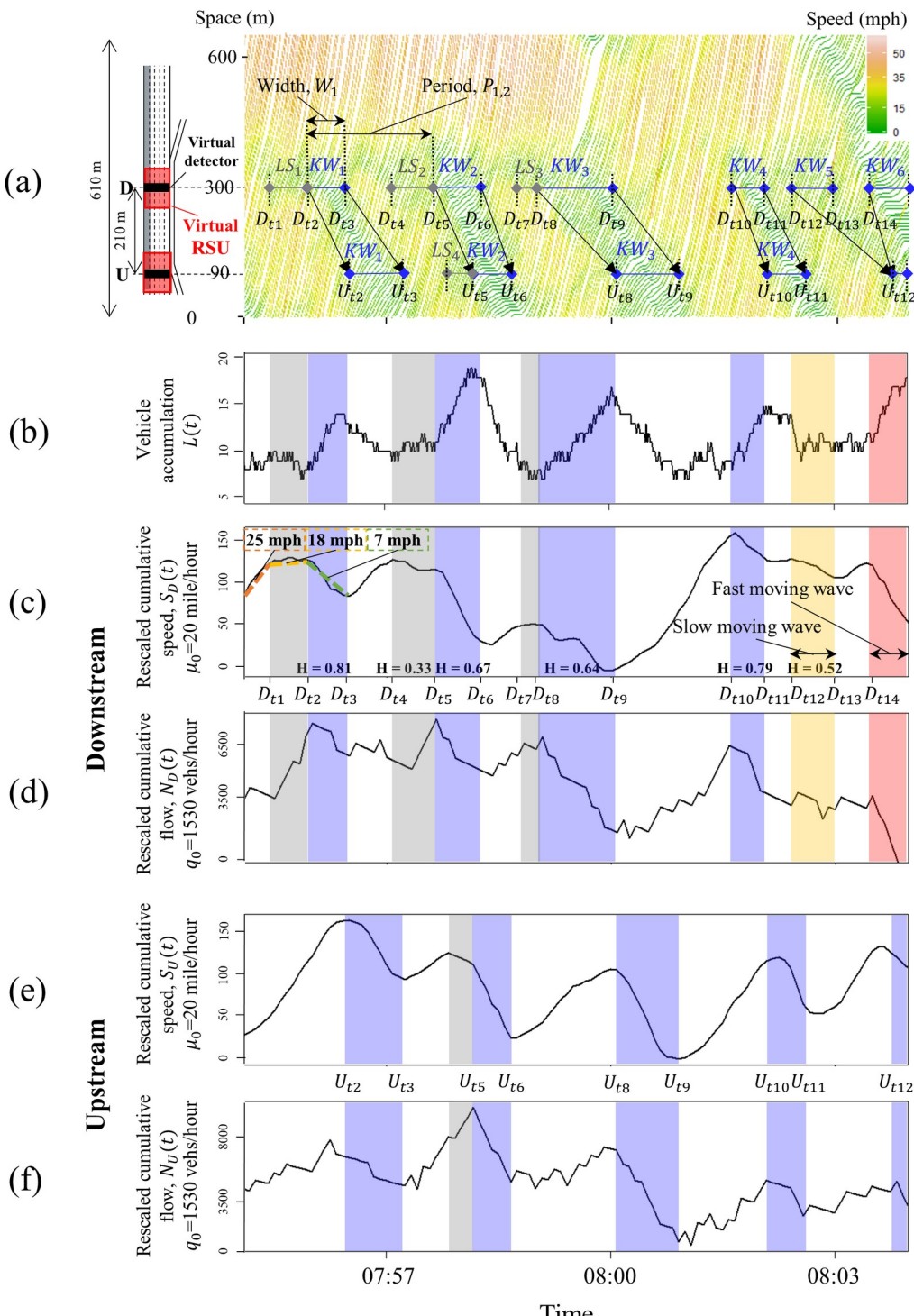

**Fig 1.** Observed features of kinematic waves within the lifecycle of traffic oscillation: (a) in trajectory data, (b) in vehicle accumulation between upstream and downstream, (c, d) rescaled cumulative flow and the downstream speed in 5-second aggregated data, and (e, f) rescaled cumulative upstream flow and speed in 5-second aggregated data.

1E mark the changes in speed observed at D and U, while Fig 1D and 1F show the changes in flow departing D and U.

The vertical grey bars in Fig 1B–1D mark the periods when localized speed reductions, i.e., $LS_1$, $LS_2$, and $LS_3$, were observed at D. The vertical grey bars shown in Fig 1E and 1F identify the time when $LS_4$ was observed at U. The vertical blue bars in Fig 1B–1F show the widths of $KW_1$ to $KW_4$, and they were observed at both D and U. The vertical orange and red bars shown in Fig 1B–1D represent the widths of $KW_5$ and $KW_6$ observed at D, and they merged before arriving at U. The widths of these vertical bars were determined by analyzing the changes in $L(t)$ together with the changes in $S(t)$ and $N(t)$ with respect to time. If the changes in $L(t)$ occurred with the simultaneously or asynchronously changes in $S(t)$ and $N(t)$, the changes observed at D are determined to be LS or KW. Although $L(t)$ has been identified as the most important parameter for detecting traffic congestion [1], it cannot be obtained directly in CVT before full-connectivity without interference. Also, the $S(t)$ and $N(t)$ require the $\mu_0$ and $q_0$, which should be selected manually after visual inspection [1, 2]. Therefore, we only used $L(t)$, $S(t)$, and $N(t)$ as reference values to differentiate the true KWs from LSs. Instead, the aggregated flow and speed, which are information that was readily available, were used as input for the proposed method.

$S(t)$ can pinpoint the times at which the changes in speed at U and D occurred. As an example, Fig 1C shows the reduction in speed at D from 25 mph to 18 mph at $D_{t1}$, followed by an additional reduction in speed to 7 mph at $D_{t2}$. The speed at D recovered at $D_{t3}$, followed by two notable reductions in the speeds observed at $D_{t4}$ and $D_{t5}$. These times mark the start of either LS or KW and become the candidates to be further evaluated by our proposed method. The backward-moving KWs, which signal changes in flow while traffic is congested, typically simultaneously accompany the changes in speed [27, 28]. However, as shown in Fig 1C and 1D, the changes in speed caused by LS did not accompany the changes in flow at the same time, so the monitoring speed alone can cause bias in estimating the locations and times of the KWs that originated near the downstream measurement location.

Monitoring changes in flow alone may result in false-positive values due to its fluctuation, as shown in Fig 1D. When $KW_1$ through $KW_4$ arrived at U, the changes in $S(t)$ and $N(t)$ occurred simultaneously. (See Fig 1E and 1F). In addition to the existence of LS, a slow-moving KW (see $KW_5$) merging with a fast-moving kinematic wave (see $KW_6$) can hinder adequate monitoring of the propagation of the wave. To address this potential bias, the authors developed a method for monitoring the generation and propagation of the KWs that considers asynchronous changes in $S(t)$ and $N(t)$. The following section provides a detailed description of the proposed method and evaluates its performance based on empirical data [23].

## Data preparation

While the conventional dual loop detectors collect the flow and speed from specific roadway locations, the CV can collect the flow and speed wherever the CV travels. The black boxes on the left side of Fig 1A show the location of the virtual detector, which mimics the conventional dual-loop detectors that consist of pairs of detectors spaced about 3 meters apart. They report the average traffic flow and speed data observed at fixed time periods, $T_L$. The time period of 30 seconds [29] typically is used for the aggregation period. In this study, both 30-second and 5-second aggregated data were used to evaluate the additional information that can be gleaned from using high-resolution data.

CV generates the trajectory of itself, which can be spatiotemporally processed further to estimate the vehicle count and speed data at a specific location. Assuming a 100% penetration rate, the vehicle trajectory data shown in Fig 1A can be considered as CV data. The two red boxes on the left side of Fig 1A show the location of the virtual RSU, which collects the traffic data transmitted by the OBU of CV at predetermined intervals; thus the traffic flow and speed

data near U and D can be estimated by different values of the time period, $T_{cv}$, and section length, $L_{cv}$. Decreasing the resolution of the data resolution (i.e., increasing the length of the section and the duration of the aggregation) can reduce the random fluctuations in the data at the cost of blurring the attributes of the KWs that are to be detected. Increasing the resolution of the data allows us to observe when the changes in the traffic condition occur at a finer time resolution; however, it is likely to be contaminated with random fluctuations. Elfar et al. [10] suggested dividing $L_{cv}$ into 60-m sections to track the KWs in the NGSIM trajectories, and we used those suggestions for representing the CV data. The $T_{cv}$ was set to 5 seconds to represent the high-resolution data, and it was set to 30 seconds to compare with the data provided by the virtual loop detector. The aggregated flow and speed for each section were calculated by Edie's definition as in Eqs (3) and (4) [30].

$$aggregated\ speed = \frac{\sum_i x_i}{\sum_i t_i} \tag{3}$$

$$aggregated\ flow = \frac{\sum_i x_i}{|A|} \tag{4}$$

where $x_i$ is the distance traveled by the $i^{th}$ vehicle in the time-space region, $t_i$ is the time spent by the $i^{th}$ vehicle in the time-space region, and $|A|$ is the area of the time-space region.

## Methods

The proposed method removes the random fluctuations and LS by sequentially comparing their features with those of the KWs. First, the observed data were decomposed using ensemble empirical mode decomposition (EEMD) [31] to filter out the temporal random fluctuation. This technique can be applied to both stationary and non-stationary data. Abrupt changes in the filtered data could have been caused by either LS, KW, or a remaining random fluctuation. Our proposed method differentiated them by comparing the similarity of the denoised speed data observed at U and D during the time periods spanning the abrupt changes. The similarity was quantified using a cross-correlation technique. Then, the resulting KW candidates were evaluated by the logistic regression model taking the asynchronous changes in the traffic flow and speed into consideration.

### Filtering out temporal random fluctuations using the EEMD

Previous studies used STFT and WT [17–19] to detect the abrupt changes accompanied by KWs; however, such methods can produce both false positives and false negatives due to the random fluctuations in the data. To filter out the random fluctuations in the non-stationary data [32–34], in this study, we applied EEMD to the flow and speed data. EEMD [31] adds the white noise to existing data, $x(t)$, prior to decomposing $x(t)$ into a set of $k$ intrinsic mode functions (IMFs), $c_i(t)$, and residue, $R(t)$, (See Eq (5)) via the sifting process. The IMFs represent the marked fluctuations in local frequency (i.e., the local period). The white noise in EEMD was added to prevent mode mixing (i.e., two signals of different frequencies coexisting in a single IMF).

$$x(t) = \sum_{i=1}^{k} c_i(t) + R(t) \tag{5}$$

The amplitude of the added white noise, $A_n$, (see Eq (6)) was set as 0.2 × the standard deviation

of the original data to avoid causing the pronounced peaks in the data to be altered, yet it was big enough to prevent mode mixing [31]. A finite, noise-added signal was decomposed into IMFs for the number of ensemble trials, $N_t$, and, when a sufficiently large number of trials were conducted, only persistent parts of the signals survived in the averaging process [31]. The two parameters, $A_n$ and $N_t$, were determined by comparing them with the difference between the input signal and the corresponding sum of IMFs, $\varepsilon_n$. (See Eq (6)) In this study, the $N_t$ was set as 1000 ($\varepsilon_n$ = 0.006 × the standard deviation of the data), which should be enough to average out the added noise [31].

$$\varepsilon_n = A_n / \sqrt{N_t} \qquad (6)$$

Fig 2B shows the result of applying the EEMD procedure to the 5-second aggregated data shown in Fig 2A. The Nyquist sampling theorem [35] states that the sample rate should be at least twice the rate of the highest frequency of the information to be extracted to capture all of the information from a continuous signal. The EEMD procedure can be applied directly to 5-second aggregated data since the inter-arrival times (i.e., oscillation periods) of the KW at the study site were significantly higher than 5 seconds. However, since the 5-second aggregated data typically would not be available in practice, the 5-second were aggregated further to 30 seconds, as shown in Fig 2C.

The 30-second aggregated data were oversampled (i.e., sampling a signal with a sampling frequency significantly higher than the target information) every 5 seconds before applying EEMD to extract the information embedded at a higher frequency. In this study, the oversampling was conducted using linear interpolation. The results of applying EEMD to the 30-second aggregated data and the oversampled 30-second aggregated data at 5 seconds (i.e., 30-second oversampled data) are shown in Fig 2D and 2E, respectively.

The noise in the present study was filtered by comparing the dominant period (DP) of IMFs computed using Hilbert spectral analysis [31] with an empirically-observed oscillation period of KW. Fig 2F shows the DP of IMPs extracted from 5-second aggregated, 30-second aggregated, and 30-second oversampled data. The DP values of different IMFs for the 5-second aggregated and 30-second oversampled data were similar, while the DP of the 30-second aggregated data showed different patterns. Previous studies reported oscillation periods of 2 to 15 minutes [17–19], and the investigation of the observed features of the KW at the study site also showed 2 minutes of oscillation periods. Based on these empirical findings, IMFs with DPs less than 2 minutes were regarded as noise. (See the black horizontal dotted line in Fig 2F.)

Filtering out IMFs based on DP can be explained with the aid of the energy of each IMF's contribution to $x(t)$. Fig 2G shows cumulative percent energy (PE) of each of the IMFs extracted from speed data that were obtained from three different levels of aggregation, and their corresponding dominant periods (DP) are shown in Fig 2F. The energy of each of the IMFs, $E_i$, can be defined as shown in Eq (7) [32–34]:

$$E_i = \sum_{i=1}^{N} \left[ c_i(t) \right]^2 \qquad (7)$$

where the $N$ is the number of data points in the $i^{\text{th}}$ IMF, $c_i(t)$. The orange lines in Fig 2F and 2G show the DP and PE of the IMFs obtained from the 30-second aggregated data. Only $IMF_1$ (see the orange vertical lines in Fig 2F) had a DP value less than 2 minutes, and it can be classified as noise. In this case, the PE of $IMF_1$ consisted of about 66 percent of the total energy, indicating that $IMF_1$ was not decomposed into noise and a relevant signal due to the low sampling rate. Removing even the first IMF alone can distort the signal markedly. In other words, filtering $IMF_1$ can remove the noise, but it also can distort the information in the signal significantly, as

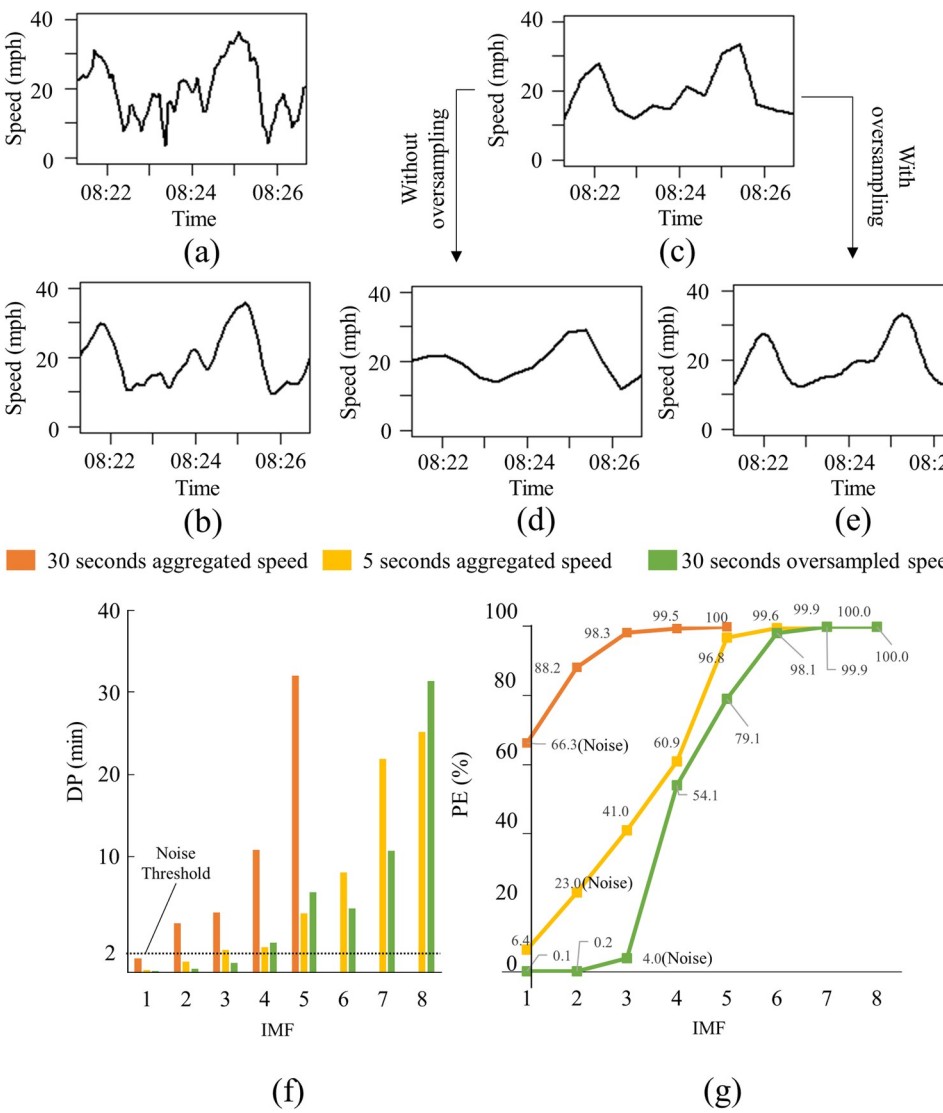

**Fig 2.** Comparison of the results of denoising each type of data: (a) 5-second aggregated data, (b) 5-second denoised data, (c) 30-second aggregated data, (d) 30-second denoised data, (e) 30-second oversampled and denoised data, (f) dominant period of IMFs, and (g) cumulative percentage energy obtained from the results of EEMD.

shown in Fig 2D. The yellow and green lines in Fig 2F and 2G show the DP and PE of the IMFs obtained from 5-second aggregated, and 30-second oversampled data, respectively. The $IMF_1$ and $IMF_2$ of the 5-second aggregated data have DP values less than 2 minutes (see the yellow vertical bars in Fig 2F), and they consisted of 23 percent PE. The $IMF_1$, $IMF_2$, and $IMF_3$ of the 30-second oversampled data reported a DP of less than 2 minutes. The combined PE of these three IMFs was only 4 percent of total PE. The low PE of the noise IMFs of the 5-second aggregated and 30-second oversampled data indicated that the signals in these IMFs likely were noise and could be filtered. Notice the differences in the DP and PE of IMFs obtained from the 5-second and 30-second aggregated data. When aggregating the 5-second data with the 30-second, information is lost about the signals that had periods shorter than 30 seconds.

Fig 2B and 2E show the 5-second aggregated data and the 30-second oversampled data, respectively, with the noise filtered. Compared with Fig 2D and 2E more closely resembles

Fig 2B, which was obtained by denoising the 5-second aggregated data. Although oversampling cannot recover the information lost from aggregation, it helps extract the information embedded in the high frequency region by linearly interpolating the signals lost during the aggregation. The proposed method, which includes the EEMD and the subsequent filtering method, was conducted with both the 5-second aggregated data and the 30-second over-sampled data.

Fig 3A shows the 5-second aggregated speed data collected from a virtual loop detector, and Fig 3B shows the corresponding denoised data. Fig 3C and 3E show the speed data collected from a virtual RSU located in D under full (i.e., 100% penetration rate) and partial connectivity (i.e., 30% penetration rate). Note that while the patterns observed in Fig 3C and 3E are markedly different, their corresponding denoised data display similar patterns. Compared with the raw loop detector data in Fig 3A, the raw CV data from the full connectivity in Fig 3B remove some minor fluctuations by reflecting the spatial aggregation along the 60-m section. Also, the raw CV data in Fig 3C and 3E indicate that the CV data from partial connectivity are more volatile and noisy than the CV data from full connectivity. Compared with the raw data, the denoised data reduce the difference between the loop detector data and the CV data from full and partial connectivity, respectively. These results imply that the EEMD shows strong promise that it can filter out random fluctuations embedded in each set of data and extract only the significant signals. The candidate KWs were selected based on the denoised data, and those candidates were evaluated further using the cross-correlation and the logistic regression model described in the following sections.

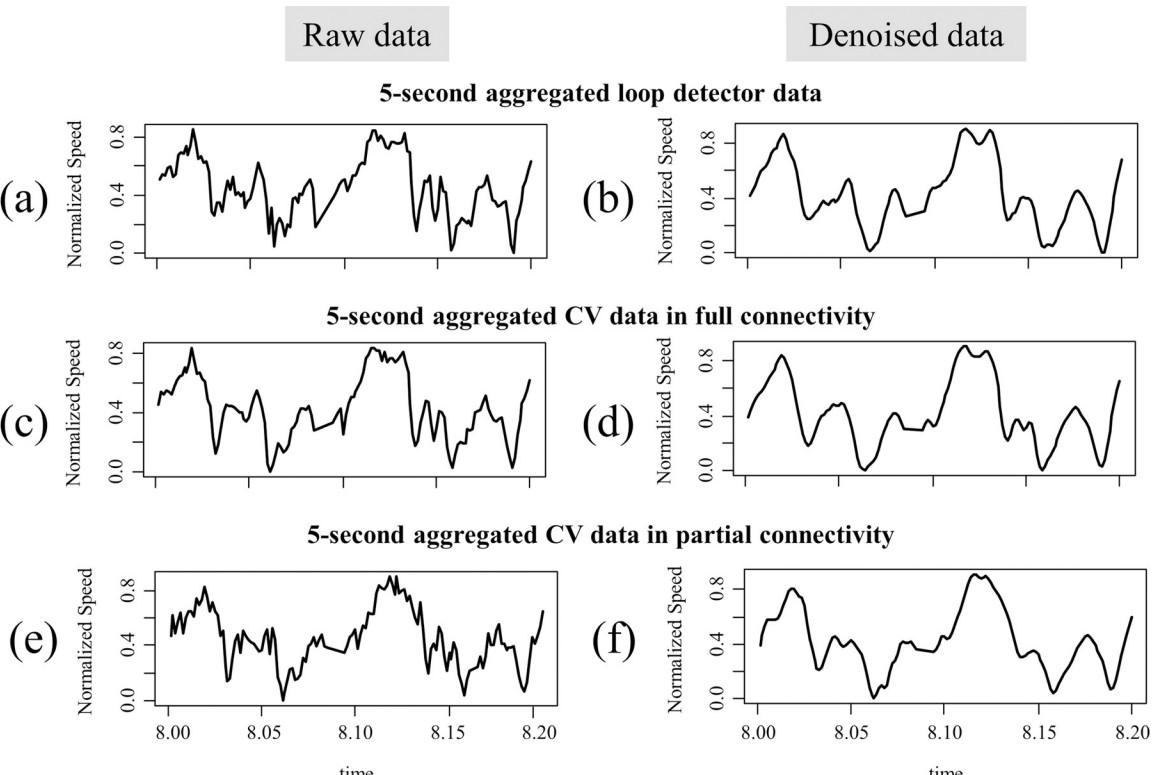

**Fig 3.** Comparison of raw and denoised 5-second aggregated speed data: (a) 5-second aggregated loop detector data and (b) corresponding denoised data; (c) 5-second aggregated CV data in full-connectivity and (d) corresponding denoised data; (e) 5-second aggregated CV data in partial-connectivity and (f) corresponding denoised data.

## Filtering out spatial random fluctuations using cross-correlation

The speed and flow relation across different sites can be markedly different [36]; however, the relative impact of KW on the changes in speed can remain comparable. Thus, we used normalized speed and flow in the range of zero to one as input to our model. Our model identifies the abrupt changes in the normalized speed, i.e., valleys in the time-series data at the downstream location, as shown in the vertical dotted line of Fig 4A. The red dotted line in Fig 4A shows one of the candidate KWs labeled as $V_{D1}$. The data within the window, $W_1$ spanning $V_{D1}$ were compared with the data observed at an upstream location within the same window and a varying time lag, $\Delta l$ as in Fig 4B: $\Delta l$ reflects the time for the KW to travel to the upstream location. The time lag ranged from 20 to 120 seconds, considering that the speed of KW can range from 6.3 km/h (3.9 mph) to 37.8 km/h (23.4 mph) [2, 3].

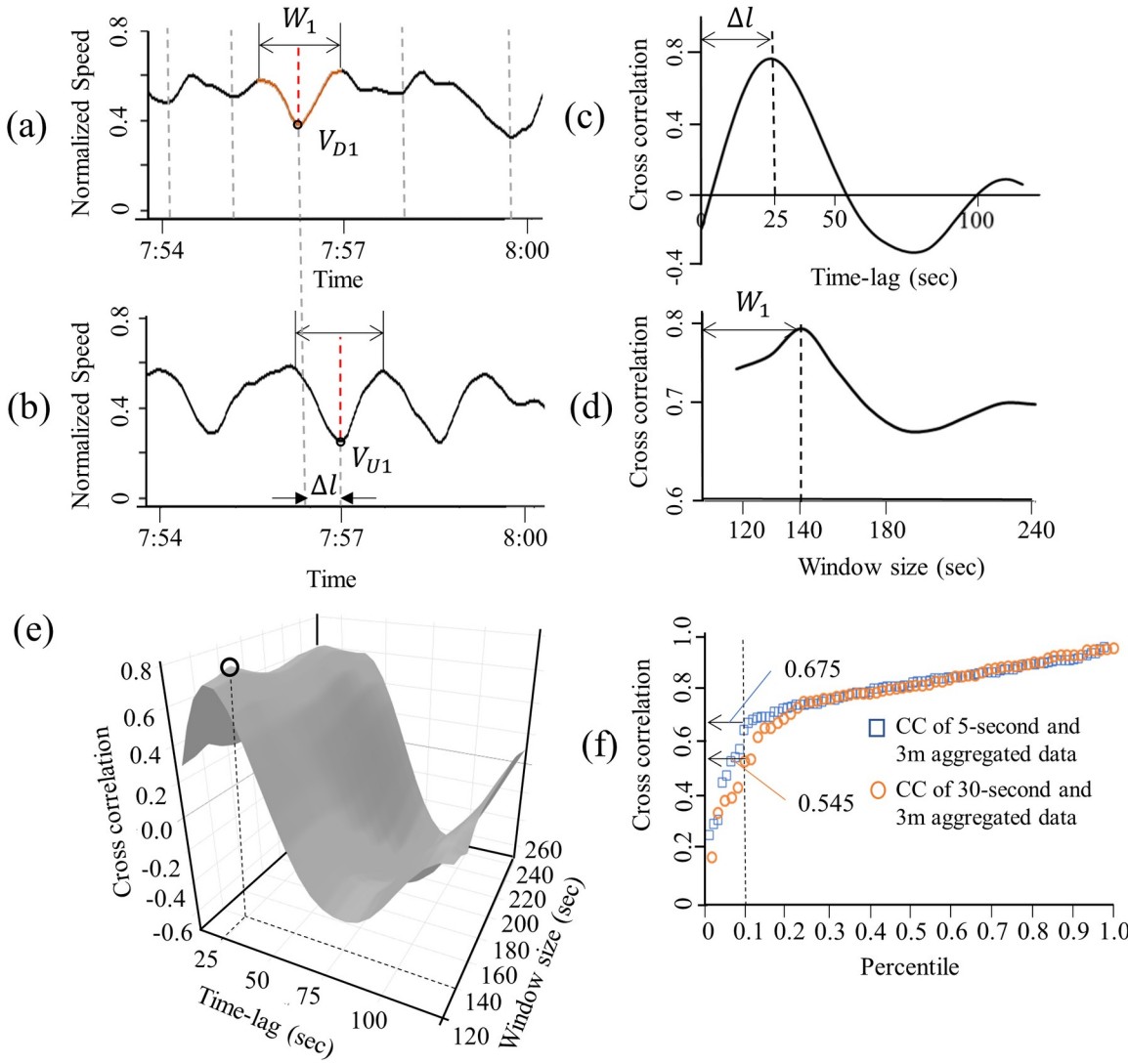

**Fig 4.** Spatial filtering for selecting candidate KWs including KW and LS, by monitoring the (a) downstream and (b) upstream normalized speed. The changes of CC according to (c) Time-lag between upstream and downstream, (d) Window size of CC, and (e) Combination of time-lag and window size. (f) Distribution of the maximum CC of each candidate KW at optimal time-lag and window size.

To match the candidate KW with the corresponding propagated valleys, the similarity between the upstream and downstream data within the window is computed using cross-correlation (CC) [20]. The size of the window ranged from two to six minutes to capture the sequential and merged KW. (See the width of KW in Fig 1A.) Arbitrarily increasing $W_1$ can distort the CC due to the influence of adjacent candidates other than $V_{D1}$. The length of $W_i$ was set to reflect a typical width of KW. Fig 4C shows the changes in CC of $W_1$ in the vicinity of $V_{D1}$ with respect to the changes $\Delta l$. Once the optimal $\Delta l$ was found, $W_1$was varied to test whether CC could be increased further by extending the length of $W_1$. (See Fig 4D) Fig 4E shows that the maximum value of CC, i.e., 0.78, was observed when $W_1$ was 140 seconds and $\Delta l$ was 25 seconds. The maximum correlation of each candidate KW was estimated in the same manner. Fig 4F shows the distribution of the maximum CC of each candidate KW at the study site. The significant reduction in CC among the bottom ten percentile indicates that the changes in speed observed at D and U are likely to be unrelated fluctuations. Since the purpose of this step is to reduce false positives, candidate KWs with CC values less than the 10th percentile were not considered in the logistic regression in the next step. This procedure was applied to data across four lanes at the study site, and a logistic regression model that classified the KWs from candidates was estimated and evaluated irrespective of the characteristics of the lane.

## Identifying the KW using the logistic regression model

Typically, KW accompanies changes in flow and speed simultaneously while LS and random fluctuation cause asynchronous changes in the flow and speed. Eq (4) and Fig 5 show how KW can be detected using such an attribute. $\vec{F}_{N1,\Delta t}$, $\vec{F}_{N2,\Delta t}$, and $\vec{F}_{N3,\Delta t}$ show the changes in normalized flow observed within $\Delta t$ (See the left axis in Fig 5) and $\vec{F}_{N1,\Delta t}$ and $\vec{F}_{N3,\Delta t}$ represent the decrease and increase of the normalized flow, respectively, while $\vec{F}_{N2,\Delta t}$, shows no changes in flow within the time interval. $\vec{S}_{1,\Delta t}$ shows the reduction in normalized speed that was observed during the same period (See the right axis in Fig 5). When congested traffic conditions exist, KW accompanies a decrease in flow together with a decrease in speed [27, 28]. When $\vec{F}_{N1,\Delta t}$ is observed with $\vec{S}_{1,\Delta t}$, it would have a higher chance of KW than when $\vec{F}_{N2,\Delta t}$, and $\vec{F}_{N3,\Delta t}$ are observed with $\vec{S}_{1,\Delta t}$. The proposed model measures the likelihood of the candidate KW being true KW by measuring the norm of a vector that is orthogonal to $\vec{S}$ and connected to $\vec{F}$. A high value of $\vec{H}$ would indicate that pronounced changes in flow accompanied the changes in speed. Conversely, low values of $\vec{H}$ would indicate that the changes in speed were not accompanied by significant changes in flow. $\vec{H}$ is the difference between $F_i$ and $F_i$ projected onto $S_i$, and it used as a parameter to filter out the changes in speed that did not accompany significant changes in flow. Eq (8) shows how $\vec{H}$ is defined numerically.

$$H = \| \vec{F_N} - \vec{S} \, (\vec{S}^T \vec{S})^{-1} \, \vec{S}^T \vec{F_N} \| \tag{8}$$

where $\vec{F_N} = -\left(t_i, flow_{t+1} - flow_t\right)$ and $\vec{S} = \left(t_i, speed_{t+1} - speed_t\right)$. Logistic regression models have been used by previous studies for analyzing categorical output data [37, 38], and advanced models considering characteristics of data such as spatial and temporal correlation also have been proposed [39, 40]. This study developed the logistic regression model that includes $H$ and the level of congestion, $C$, which account for the effects of vehicle spacing related to the generation of waves from instability and lane changes in congested traffic [28]. The value of $C$ was computed as a normalized speed at the valley of candidate KW. (See $V_{D1}$ in Fig 4A).

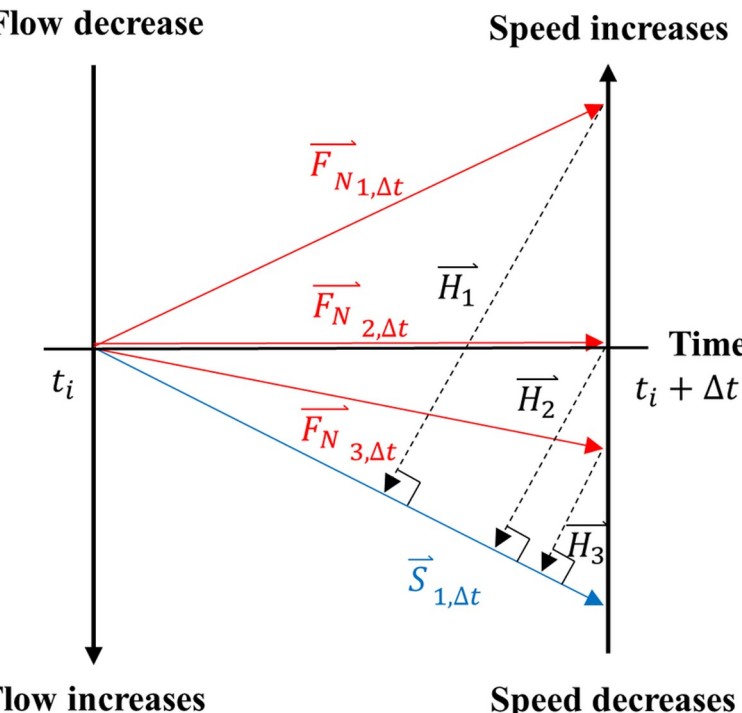

**Fig 5. Geometric features of *H* reflecting asynchronous changes in speed and flow.**

The speed and flow relation across different lanes would be different due to various factors such as the traffic composition and proximity to on-off ramps [36]. To consider the effects of lane-specific characteristics, we include the random effects parameter in the mixed effect logistic regression (MELR) model [41, 42]. The proposed MELR models assume that the probability of being KW within lanes may be correlated. The probability that the candidate is the true KW given $\eta$ in the MELR model is described as in Eqs (9) and (10).

$$\eta = X\boldsymbol{\beta} + Z\boldsymbol{u} \tag{9}$$

$$p(Y = 1|\eta) = \frac{exp(\eta)}{1 + exp(\eta)} \tag{10}$$

where the input data $X$ is a $N\times3$ matrix of the 2 independent variables (i.e., $H$ and $C$) and intercept; $N$ is the number of data points; $\boldsymbol{\beta}$ is $3\times1$ vector of the parameters for fixed effects; $Z$ is $N\times4$ matrix for 4 different lanes; $\boldsymbol{u}$ is $4\times1$ vector of the random effects. The components of $\boldsymbol{\beta}$ are respectively estimated parameters for the $C$, $H$, and intercept (i.e., $\beta_1$, $\beta_2$, and $\beta_0$), and $\boldsymbol{u}$ is distributed as normal with zero mean and variance $V$ that is the variance-covariance matrix of the random effects. Since we includes the random effects as deviations from the fixed intercepts effects ($\beta_0$), $V$ is the variance of random intercept effects. The MELR model is estimated based on the Laplace approximation to the log-likelihood. More details about computational approach for model estimation are presented in [42]. We performed the repeated 5-fold cross-validation 10 times to evaluate the performance of MELR model. The candidate KWs were selected as a KW when the $p(Y = 1|\eta) > 0.5$ (Eq (10)), and the results of applying the proposed method to the study site are reported in the following section.

## Findings

Fig 6 shows how the candidate KWs were filtered using EEMD, CC, and the logistic regression. The data were based on the CV in full connectivity. The white circles in Fig 6A mark when the changes in speed were observed at D, and they first were filtered using EEMD, as shown in Fig 6B. Then, the candidate KW was evaluated further using CC, and Fig 6C shows the candidate KW with CC greater than 0.545, which eliminates the bottom $10^{th}$ percentile of the candidate KW in terms of CC. Fig 6D shows the performance of the model in terms of false-positives (FPs), false negatives (FNs), true positives (TPs), and true negatives (TNs) based on the true KW shown in Fig 6E. The red boxes in Fig 6E mark the KW that emanated further down-stream of D, displaying a better-defined width of KW and a stabilized speed. The blue boxes in Fig 6E mark the KW that emanated near D. Most of the FPs and FNs occurred in detecting KWs that emanated near D. This could be the result of KWs that were not stabilized until they traveled further upstream, and it also could indicate the increase in the amplitude of KW as it was propagated [3, 16, 43].

Fig 7 shows the effects of the level-of-congestion ($C$) and asynchronous changes in speed and flow ($H$) on detecting KW in the MELR excluding random effects. Fig 7A and 7B show how the $p(Y = 1)$ changes with respect to the changes in $C$ and $H$ in the 5-second aggregated data collected from a CV in full and partial connectivity, respectively. When the threshold of $p(Y = 1)$ is set to be 0.8, the black region marks the combination of $C$ and $H$ that will identify the candidate KW as the true KW, and the other region marks the combination of C and H that will mark the KW as a false KW. The result of the analysis can change depending on the threshold value. The grey regions in Fig 7A and 7B mark the combination of $C$ and $H$ that can be identified as true KW or false KW when the threshold value ranges from 0.2 to 0.8. The number of gray regions of Fig 7A and 7B are comparable and less than 15 among 100 data points, indicating that the sensitivity of the results are low to different values of the threshold and connectivity of the data. To be conservative in our analysis, we used $p(Y = 1)$ set to be 0.5.

Table 1 shows the estimated parameters of the MELR model using different types of data. (See the first to third column of Table 1.) The statistical significances of $C$ and $H$ at different p-values also are included in Table 1. The sign of $H$ is the same as expected, i.e., the greater the value of $H$, the higher the likelihood becomes of the candidate KW being the true KW. The negative coefficient of $C$ indicates that the KW is more likely to occur where the level-of-congestion increases, as expected. The significances of $C$ and $H$ are more prominent in the 5-second aggregated data than in the 30-second aggregated data. These results show that temporal aggregation to 30 seconds would remove significant information from the data, which would affect the likelihood of candidates being KW. The standard deviation (SD) of the random effects caused by lane is also estimated. The SD of random effects in fully-connected 5-second and 30-second aggregated data are respectively 2.79 and 3.05, which indicate that there are significant lane effects among KWs. Those effects may be attributed to some unobserved factors such as the proportion of heavy vehicles and driving behavior in each lane. In both 5-second and 30-second data, the SD of random effects decreases in partial-connected data and loop detector data, compared with the fully connected data. The results indicate that a reduction in the quality of the data caused by sample rate and spatial coverage causes underestimation of lane-specific effects.

Table 2 shows the detection performance of the proposed method in various data conditions, such as the type, resolution, and connectivity of the data. The fourth column of Table 2 shows the total number of initial KW candidates observed at different data conditions. The objective of the proposed method was to improve the performance of the model by removing fluctuations that were not related to KW. The remaining columns of Table 2 report the

**Fig 6.** Step-by-step results of the proposed method applied to the 5-second aggregated data collected from a CV in a fully-connected environment: (a) Raw speed data, (b) Denoising using EEMD, (c) Spatial filtering using CC, (d) Identifying KW using logistic regression, and (e) Traffic conditions in the time-space diagram.

detection performance considering all errors that occurred in the sequential processes, including data aggregation, temporal filtering, spatial filtering, and logistic regression. Since the

## (a) $Pr(Y = 1 \mid C, H)$ in full-connectivity

| | < 0.2 | 0.2 ~ 0.8 | > 0.8 |
|---|---|---|---|

| H | | | | | | | | | | |
|---|---|---|---|---|---|---|---|---|---|---|
| 1 | 0.00 | 0.00 | 0.00 | 0.00 | 0.00 | 0.00 | 0.00 | 0.00 | 0.00 | 0.01 |
| 0.9 | 0.00 | 0.00 | 0.00 | 0.00 | 0.00 | 0.00 | 0.00 | 0.01 | 0.02 | 0.05 |
| 0.8 | 0.00 | 0.00 | 0.00 | 0.00 | 0.00 | 0.01 | 0.02 | 0.05 | 0.13 | 0.29 |
| 0.7 | 0.00 | 0.00 | 0.00 | 0.01 | 0.02 | 0.06 | 0.14 | 0.31 | 0.54 | 0.76 |
| 0.6 | 0.00 | 0.01 | 0.03 | 0.07 | 0.16 | 0.33 | 0.56 | 0.77 | 0.90 | 0.96 |
| 0.5 | 0.03 | 0.07 | 0.17 | 0.35 | 0.58 | 0.79 | 0.91 | 0.96 | 0.99 | 0.99 |
| 0.4 | 0.18 | 0.37 | 0.61 | 0.80 | 0.92 | 0.97 | 0.99 | 1.00 | 1.00 | 1.00 |
| 0.3 | 0.63 | 0.82 | 0.92 | 0.97 | 0.99 | 1.00 | 1.00 | 1.00 | 1.00 | 1.00 |
| 0.2 | 0.93 | 0.97 | 0.99 | 1.00 | 1.00 | 1.00 | 1.00 | 1.00 | 1.00 | 1.00 |
| 0.1 | 0.99 | 1.00 | 1.00 | 1.00 | 1.00 | 1.00 | 1.00 | 1.00 | 1.00 | 1.00 |
| | 0.1 | 0.2 | 0.3 | 0.4 | 0.5 | 0.6 | 0.7 | 0.8 | 0.9 | 1 *C* |

## (b) $Pr(Y = 1 \mid C, H)$ in partial-connectivity

| H | | | | | | | | | | |
|---|---|---|---|---|---|---|---|---|---|---|
| 1 | 0.00 | 0.00 | 0.00 | 0.00 | 0.00 | 0.00 | 0.00 | 0.00 | 0.00 | 0.00 |
| 0.9 | 0.00 | 0.00 | 0.00 | 0.00 | 0.00 | 0.00 | 0.00 | 0.00 | 0.00 | 0.01 |
| 0.8 | 0.00 | 0.00 | 0.00 | 0.00 | 0.00 | 0.00 | 0.00 | 0.02 | 0.05 | 0.13 |
| 0.7 | 0.00 | 0.00 | 0.00 | 0.00 | 0.01 | 0.02 | 0.06 | 0.16 | 0.37 | 0.64 |
| 0.6 | 0.00 | 0.00 | 0.01 | 0.02 | 0.07 | 0.20 | 0.42 | 0.69 | 0.88 | 0.96 |
| 0.5 | 0.01 | 0.03 | 0.09 | 0.23 | 0.48 | 0.74 | 0.90 | 0.96 | 0.99 | 1.00 |
| 0.4 | 0.11 | 0.27 | 0.53 | 0.78 | 0.92 | 0.97 | 0.99 | 1.00 | 1.00 | 1.00 |
| 0.3 | 0.59 | 0.82 | 0.93 | 0.98 | 0.99 | 1.00 | 1.00 | 1.00 | 1.00 | 1.00 |
| 0.2 | 0.94 | 0.98 | 0.99 | 1.00 | 1.00 | 1.00 | 1.00 | 1.00 | 1.00 | 1.00 |
| 0.1 | 1.00 | 1.00 | 1.00 | 1.00 | 1.00 | 1.00 | 1.00 | 1.00 | 1.00 | 1.00 |
| | 0.1 | 0.2 | 0.3 | 0.4 | 0.5 | 0.6 | 0.7 | 0.8 | 0.9 | 1 *C* |

**Fig 7.** Sensitivity analysis for probability to be KW with *C* and *H* for the 5-second aggregated data collected from a CV in (a) full connectivity and (b) partial connectivity.

logistic regression model was evaluated by repeated 5-fold cross-validation with 10 trials, we used the average values of TP, TN, FP, and FN obtained in 10 trials. While the number of TPs (i.e., the number of KWs in the study site) was the same for all kinds of data, the number of TNs (i.e., LS and random fluctuations in the raw data) was different according to the conditions of the data.

In the application of detecting KW, the low precision, i.e., the ratio between TP and the sum of TP and FP, causes unnecessary traffic control in a situation where KW does not occur, thus causing inefficient traffic flow, while low recall, i.e., the ratio between TP and the sum of TP and FN, causes congestion because it cannot cope with the KW generated downstream. Both precision and recall have important implications in real-time traffic control, but changing the threshold of the MELR model can increase either precision or recall and decrease the

**Table 1. Results of the MELR model from candidate KWs of four lanes.**

| Resolution | Type | Connectivity | Intercept | | C | | H | |
|---|---|---|---|---|---|---|---|---|
| | | | Fixed effects | Standard deviation of random effects | | | | |
| 5-second aggregated | CV | Full | 5.66 \| * | 2.79 | -20.36 \| *** | | 9.73 \| * | |
| | | Partial | 6.69 \| ** | 1.23 | -24.93 \| *** | | 11.35 \| *** | |
| | Loop detector | - | 4.80 \| ** | 0.94 | -16.25 \| *** | | 5.00 \| * | |
| 30-second aggregated | CV | Full | 6.80 \| *** | 3.05 | -25.35 \| *** | | 17.61 \| ** | |
| | | Partial | 3.38 \| * | 2.43 | -8.81 \| ** | | 5.83 \| * | |
| | Loop detector | - | 6.49 \| ** | 1.52 | -16.67 \| ** | | 6.41 | |

*Note*: Parameters that are significant at the *p*-5% level:

* $p < .10$;

** $p < .05$;

*** $p < .01$.

other. We used the F1 score, which is the harmonic mean of precision and recall, as a performance measure to account for both precision and recall. The F1 scores are reported in the last column of Table 2.

Regarding the 5-second aggregated data, the proposed method provided 95.6% precision, 88.8% recall, and had a 92.5% F1 score when the data were collected from a CV in full connectivity. The F1 score decreased slightly in the data collected from partial connectivity to 91.1%. This result shows the strong promise of the proposed method in both the fully- and partially-connected environment. The method also shows comparable performance in the data collected from loop detector data, which have an F1 score of 90.9%.

Regarding the 30-second aggregated data, even for the data collected from a CV in full connectivity, the proposed method achieved 96.0% precision, 81.4% recall, and an F1 score of 88.1%. When the method was applied to the data collected from a CV in partial connectivity, the performance decreased as indicated by the 82.0% F1 score. These results indicate that the key features in detecting KW would be averaged out in the 30-second aggregated data even if the trajectories of all of the vehicles are used. Therefore, when designing a CVT to deal with the KW, it is important to collect the data with high temporal resolution.

Fig 8 shows the breakdown of total false detection, which consists of FN and FP from using 30-second and 5-second aggregated data from the fully-connected and partially-connected environments. The total number of false detections decreased markedly for both the full and partial CV conditions when the resolution of the data was changed from 30-second resolution to 5-second resolution. The difference in the number of false detections could have been caused by the loss of information during the aggregation of the data. In both the fully-connected and partially-connected environments, the use of the 30-second aggregated data resulted in five false detections.

**Table 2. Performance of the proposed method in detecting various data resolutions.**

| Resolution | Type | Connectivity | Number of candidates | Precision | Recall | F1 score |
|---|---|---|---|---|---|---|
| 5-second Aggregated | CV | Full | 112 | 95.6% | 88.8% | 92.5% |
| | | Partial | 115 | 92.8% | 89.1% | 91.1% |
| | Loop detector | Full | 124 | 92.2% | 87.1% | 90.9% |
| 30-second Aggregated | CV | Full | 78 | 96.0% | 81.4% | 88.1% |
| | | Partial | 79 | 92.0% | 73.9% | 82.0% |
| | Loop detector | Full | 78 | 93.4% | 75.9% | 83.7% |

Fig 8 also shows that applying EEMD during the temporal filtering resulted in only one or two false detections in all four cases. As expected, a greater number of false detections was observed during the spatial filtering in the 30-second aggregated data than in the 5-second aggregated data due to the loss of information. The greatest number of false detections occurred while finalizing the prediction using the MELR model since the about 75% to 80% of the initial candidates KW (i.e., all the valleys in Fig 6A) were classified as KW or not, using the MELR model. In the full connectivity, the number of false detections caused by the MELR was larger in the 5-second aggregated data than in the 30-second aggregated data since the number of candidate KWs in the 5-second aggregate data was much more than the number in the 30-second aggregated data. However, the difference of false detection in the MELR was less than those in the previous steps. The number of false detections in the fully-connected conditions was lower than the number in the partially-connected condition in both 30-second and 5-second resolution since, as expected, the partial connectivity also caused the loss of information. In particular, the number of false detection in 30-second aggregated data collected from partial connectivity is significantly larger than other data types. The result indicates that the traffic information for detecting KW would be drastically lost in partially connected 30-second aggregated data.

## Concluding remarks

Existing methods for detecting KW by amplifying the signal of interest using different sizes of windows (i.e., methods using STFT, WT, or SSD) or using solely speed or flow can result in high rates of false positives due to random fluctuations (i.e., noise) and the existence of LS in the data. Our proposed method uses a sequence of steps to remove random fluctuations, and it considers asynchronous changes in flow and speed in detecting KW.

The proposed method detected the KW using data collected from various data conditions, such as temporal aggregation (i.e., 5-second or 30-second), the type of data (i.e., loop detector or CV-generated data covering a 60-m section), and partial or full connectivity (i.e., 30% or 100% penetration rate). Based on the observed attributes of KWs, we identified the candidate KWs based on the changes in speed observed with respect to time. The noise in these candidate KWs was filtered out temporarily by decomposing the data into a set of IMFs. Then, the IMFs that represented random fluctuations in the data were filtered out by comparing the dominant period of IMF with the previously-known periods of the KWs. The results of denoising showed that the EEMD separated the noise in the high frequency from significant information in both

## Number of false detection

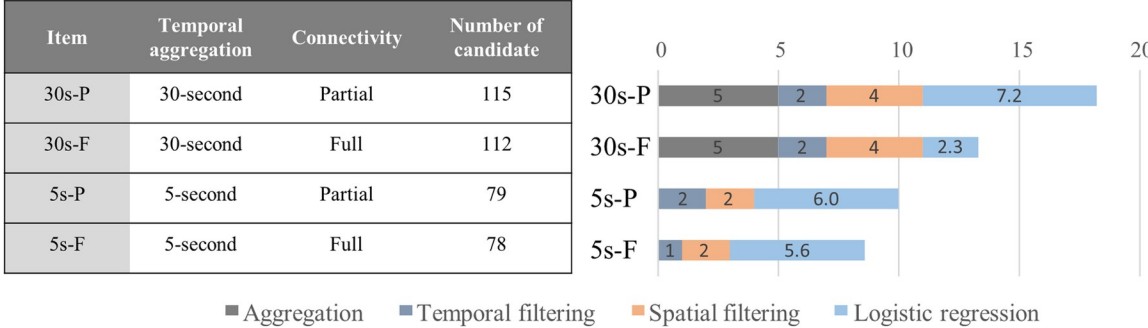

**Fig 8. Contribution of the process to false detections in the proposed method.**

the 5-second aggregated data and 30-second oversampled data. Then, the temporally-denoised data using EEMD were evaluated spatially using CC.

The CC of the denoised data over the space was evaluated to identify candidate KWs to be analyzed by the logistic regression in the subsequent procedure. The distribution of the CCs of the candidate KWs abruptly decreased at the specific percentile for both the 5-second and the 30-second aggregated data, and it was used as a threshold in selecting candidate KWs.

Then, the logistic regression model was used to determine whether or not to classify the spatiotemporally filtered data as KW. The performance of the proposed model was evaluated by applying 5-fold cross-validation ten times. One of the two parameters of the logistic regression was the normalized speed of the candidates KW (i.e., the valleys in the time-series data at the downstream location) that represents the level-of-congestion of traffic state. The other parameter reflects the degree of asynchronous changes in the denoised flow and speed.

The proposed method obtained a 92.5% F1 score in detecting KW using 5-second aggregated data from a CV in the full connectivity, and it obtained a 91.1% F1 score in partial connectivity. These findings showed strong promise for the proposed method in detecting KW that can be instrumental in developing a real-time control strategy of connected vehicles [24, 25]. Several practical implications can be drawn from the findings of this study. Most of the falsely-identified KWs were emanated near D, while the bottleneck remained active. (See Fig 6E.) This may be the result of KWs not yet becoming stable since this occurs as they travel further upstream [3, 16, 37]. Therefore, when the proposed method is applied in practice, the several RSUs collecting traffic data should be installed at specific intervals to detect the downstream's false negatives in the upstream. Performance evaluation in the various data conditions was used to investigate the impacts of the type of data, temporal aggregation, and penetration rate on detecting KW, and it revealed that high temporal resolution is an important condition of the data for detecting KW. Although data aggregation can reduce the noise of the data, the missing attributes of traffic flow is more significant in detecting KWs. Also, the proposed method well-performed 5-second aggregated data even they collected from partial connectivity and loop detector. Therefore, when designing a data collection system for CV, transportation agencies should put a lot of effort into obtaining a high temporal resolution of the data, and the random fluctuation of those data can be effectively removed by denoising methods such as EEMD.

Although the aim of the proposed method was to detect the KW from the data collected in a connected environment, we did not examine the reasons KW was generated, such as lane changes. The impact of the causative factors on detecting KW should be discussed in further analyses. The empirical evaluation showed that the proposed parsimonious model, which uses the level-of-congestion and asynchronous changes in speed and flow, effectively can identify the KWs in the traffic data denoised by the proposed spatiotemporal filtering method. However, other technique for temporal filtering [44], other statistics for spatial filtering [45], and a sophisticated classification model to identify the KW can improve the performance of the proposed method. Evaluating the performance of the model with various techniques will be the subject of future study.

Data availability is also a limitation of this study. Obtaining trajectories of all the vehicles on the road is so difficult since it is extracted from videos of several synchronized cameras mounted on top of high buildings adjacent to the roadway. To the best of our knowledge, the vehicle trajectory data of NGSIM is the only open-source data that include all trajectories of vehicles on the congested highway. Recently, unmanned aerial vehicles have been proposed to provide images above the traffic, but it is difficult to stably provide such 610 m long and 45 minutes of high-resolution video [21]. Applying the proposed method to other trajectory data that mimic the data from CVT, or to real CV data would be the necessary and direct subject of future study.

## Supporting information

**S1 Dataset. The raw data employed in this study.**
(ZIP)

## Author Contributions

**Conceptualization:** Eui-Jin Kim, Koohong Chung.

**Formal analysis:** Eui-Jin Kim, Koohong Chung.

**Funding acquisition:** Dong-Kyu Kim.

**Investigation:** Eui-Jin Kim.

**Methodology:** Eui-Jin Kim, Koohong Chung.

**Project administration:** Dong-Kyu Kim, Seung-Young Kho.

**Resources:** Dong-Kyu Kim, Seung-Young Kho.

**Supervision:** Koohong Chung.

**Validation:** Eui-Jin Kim.

**Writing – original draft:** Eui-Jin Kim.

**Writing – review & editing:** Eui-Jin Kim, Koohong Chung.

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
