## [Decision Letter · Decision Letter 0]

21 Oct 2020

PONE-D-20-30079

Spatiotemporal filtering method for detecting kinematic waves in a connected environment

PLOS ONE

Dear Dr. Chung,

Thank you for submitting your manuscript to PLOS ONE. After careful consideration, we feel that it has merit but does not fully meet PLOS ONE’s publication criteria as it currently stands. Therefore, we invite you to submit a revised version of the manuscript that addresses the points raised during the review process.

We look forward to receiving your revised manuscript.

Kind regards,

Feng Chen

Academic Editor

PLOS ONE

Journal Requirements:

"Dong-Kyu Kim was supported by the Basic Science Research Program through the National Research Foundation of Korea (NRF) funded by the Ministry of Science and ICT  No. 2020R1F1A1074395.(https://www.nrf.re.kr/eng/index)

The funders had no role in study design, data collection and analysis, decision to publish, or preparation of the manuscript.".

i) Please provide an amended statement that declares *all* the funding or sources of support (whether external or internal to your organization) received during this study, as detailed online in our guide for authors at http://journals.plos.org/plosone/s/submit-now.  Please also include the statement “There was no additional external funding received for this study.” in your updated Funding Statement.

ii) Please include your amended Funding Statement within your cover letter. We will change the online submission form on your behalf.

Reviewers' comments:

Reviewer's Responses to Questions

**Comments to the Author**

1. Is the manuscript technically sound, and do the data support the conclusions?

Reviewer #1: Yes

Reviewer #2: Yes

2. Has the statistical analysis been performed appropriately and rigorously? 

Reviewer #1: No

Reviewer #2: Yes

3. Have the authors made all data underlying the findings in their manuscript fully available?

Reviewer #1: Yes

Reviewer #2: Yes

4. Is the manuscript presented in an intelligible fashion and written in standard English?

Reviewer #1: Yes

Reviewer #2: Yes

5. Review Comments to the Author

Reviewer #1: This paper proposes a spatiotemporal filtering method for detecting kinematic waves in a connected environment. The research topic is interesting and worth of investigation. However, the research gap is not stated clearly in the Introduction or Background Section. The research gap is very important, as it implies the potential contribution of this research.

According to the Data Section, the data on vehicle trajectories are collected on June 15, 2005 from 7:50 to 8:35 A.M. They are too old, and the observation period is short. The empirical analysis conducted on the data observed in recent years and during a long period (such as a week or a month) is preferred.

The logistic regression proposed for identifying the KW is somewhat simple. As the observed data are spatially and temporally correlated, accounting for the spatial and temporal correlations in the logistic model is suggested. The following is some representative works on spatial and temporal regression models applied in the field of transportation engineering for the references:

Investigating the impacts of real-time weather conditions on freeway crash severity: A Bayesian spatial analysis. International Journal of Environmental Research and Public Health, 2020, 17(8), 2768.

Jointly modeling area-level crash rates by severity: A Bayesian multivariate random-parameters spatio-temporal Tobit regression. Transportmetrica A: Transport Science, 2019, 15(2): 1867-1884.

Their findings indicate that accommodating spatial and temporal correlations can significantly improve model fit and reduce model misspecification.

Besides, more discussions on the results and the practical implications of the findings are expected.

Reviewer #2: The topic of this paper is interesting. The methods sound. The results are meaningful and useful. There are several suggestions to improve this paper.

1. How the data of the trajectories of individual vehicles in all of the lanes along the 640-m section of the freeway were collected? The authors could refer to the following one if the video technique is used.

[1] “Using the visual intervention influence of pavement markings for rutting mitigation-part I: preliminary experiments and field tests”, International Journal of Pavement Engineering, 2019, 20(6), 734-746.

2. More references of logit models are needed. For example, the following ones.

[2]  “Investigating the Differences of Single- and Multi-vehicle Accident Probability Using Mixed Logit Model", Journal of Advanced Transportation, 2018, UNSP 2702360.

[3] Analysis of hourly crash likelihood using unbalanced panel data mixed logit model and real-time driving environmental big data. 2018, JOURNAL OF SAFETY RESEARCH. 65: 153-159.

3. The definition of the figures is not high enough.

6. PLOS authors have the option to publish the peer review history of their article (what does this mean?). If published, this will include your full peer review and any attached files.

Reviewer #1: No

Reviewer #2: No

---

## [Author Response · Author response to Decision Letter 0]

26 Nov 2020

The authors really appreciate the anonymous referees who reviewed this paper for their constructive and helpful comments. In the revised manuscripts, essential modifications were made based on comments of editor and reviewers. Some of the major changes to the revised manuscripts include applying advanced logistic regression model, detailed description of the research gaps and contributions, limitations of the data availability, and practical implications of the findings. All the revision can be tracked in the manuscripts and point-to-point responses to each reviewer comments are included in the separate files the "Response to Reviewer.

---

## [Decision Letter · Decision Letter 1]

8 Dec 2020

Spatiotemporal filtering method for detecting kinematic waves in a connected environment

PONE-D-20-30079R1

Dear Dr. Chung,

We’re pleased to inform you that your manuscript has been judged scientifically suitable for publication and will be formally accepted for publication once it meets all outstanding technical requirements.

Kind regards,

Feng Chen

Academic Editor

PLOS ONE

Additional Editor Comments (optional):

Reviewers' comments:

Reviewer's Responses to Questions

**Comments to the Author**

1. If the authors have adequately addressed your comments raised in a previous round of review and you feel that this manuscript is now acceptable for publication, you may indicate that here to bypass the “Comments to the Author” section, enter your conflict of interest statement in the “Confidential to Editor” section, and submit your "Accept" recommendation.

Reviewer #1: All comments have been addressed

Reviewer #2: All comments have been addressed

2. Is the manuscript technically sound, and do the data support the conclusions?

Reviewer #1: (No Response)

Reviewer #2: Yes

3. Has the statistical analysis been performed appropriately and rigorously? 

Reviewer #1: (No Response)

Reviewer #2: Yes

4. Have the authors made all data underlying the findings in their manuscript fully available?

Reviewer #1: (No Response)

Reviewer #2: Yes

5. Is the manuscript presented in an intelligible fashion and written in standard English?

Reviewer #1: (No Response)

Reviewer #2: Yes

6. Review Comments to the Author

Reviewer #1: (No Response)

Reviewer #2: (No Response)

7. PLOS authors have the option to publish the peer review history of their article (what does this mean?). If published, this will include your full peer review and any attached files.

Reviewer #1: No

Reviewer #2: No

---

## [Editor Report · Acceptance letter]

10 Dec 2020

PONE-D-20-30079R1 

Spatiotemporal filtering method for detecting kinematic waves in a connected environment 

Dear Dr. Chung:

I'm pleased to inform you that your manuscript has been deemed suitable for publication in PLOS ONE. Congratulations! Your manuscript is now with our production department. 

Kind regards, 

on behalf of

Dr. Feng Chen 

Academic Editor

PLOS ONE